# The sum of all parts: A multi-level exploration of racial and ethnic identity formation during emerging adulthood

**Stephanie A. Grilo** *, **Mollie R. Semler, Stephanie Rameau**

Heilbrunn Department of Population and Family Health, Columbia University Mailman School of Public Health, New York, NY, United States of America

\* sag2179@cumc.columbia.edu

## Abstract

According to the 2020 U.S. Census, the multiracial population measured at 33.8 million, nearly a 300% increase from the 2010 U.S. Census. The significant increase is due in part to improvements in measures to categorize this population. However, there is a dearth of research on the factors and processes that affect multiracial identity formation. The researchers investigated the precipitating factors to the formation of multiracial identification. Participants were recruited through social media campaigns. Hour-long in-depth interviews were conducted with 21 participants via Zoom, following an interview guide consisting of 9 categories: racial and ethnic identification, childhood and upbringing, family influence, peer engagement, health and wellbeing, experiences of discrimination, forming resilience, language, and demographics. Coding of transcripts and thematic analyses revealed that individual, interpersonal and community level influences influenced identity development differently depending on the individual's positionality within the life course. This supported the use of both the life course framework as well as the social ecological framework when examining multiracial identity development.

## Introduction

The population of adolescents and young adults who identify as multiracial, or as having parents who are of two or more racial groups, in the United States has increased significantly in the past ten years and is the fastest-growing demographic [1]. According to the 2020 U.S. Census results, the multiracial population is now measured at 33.8 million people in 2020, a nearly 300% increase from the 2010 U.S. Census result [2]. Individuals younger than 18 years of age who identify as multiracial now comprise 53% of the demographic, a 6-point increase from the 2010 results [2]. The significant increase is in part due to improvements in measures to categorize this population. Specifically, the data collection mechanisms for capturing the ethnic and racial diversity of the multiracial population have progressed to more accurately experience the self-identification of multiracial individuals.

A review of the literature on multiracial identity formation in adolescence indicates that the Ecological Framework for Understanding Multiracial Identity Development is a necessary

**Data Availability Statement:** All relevant data are within the paper.

**Funding:** Initials of the authors who received each award: SAG Grant numbers awarded to each author: (N/A) The full name of each funder:

Columbia University Irving Medical Center Grants Program for Junior Faculty Who Contribute to the Diversity Goals of the University URL of each funder website: https://provost.columbia.edu/content/junior-faculty-grants The funders had no role in study design, data collection and analysis, decision to publish, or preparation of the manuscript.

**Competing interests:** The authors have declared that no competing interests exist.

intersectional framework approach to understanding the influences on multiracial identity formation [3]. Examining multiracial identification using a multi-level conceptual framework is critical because an interdisciplinary approach provides a richer understanding of the experiences of multiracial adolescents [4]. Previous research has demonstrated a relationship between peer network behavior and the influence of peers on how multiracial youth self-identify [5]. Using this framework, four primary categories of factors influence the racial and ethnic self-identification of multiracial individuals throughout the childhood, adolescence, and adulthood stages of the life course: structural, community, and interpersonal, and individual factors. For example, the history of how whiteness as a racial and social category as been created in United States history are structural factors which are foundational to present-day racism and discrimination [6].

## Conceptual framework

The conceptual framework for this paper was drawn from the social ecological framework. The development and goal of this framework is described in full in a previous paper [3]. The social ecological framework integrates influences from the individual, interpersonal, community and structural levels. It also emphasizes that these levels interact with one another, and that interaction of factors is as important as each individual level [7]. Building on the use of the social ecological model, the conceptual framework also incorporates the life course framework. The reason for this addition is that the process of racial and ethnic identification does not happen at one stage and stay stagnant, but instead takes place over time and may be influenced by different factors at different times. In this analysis we ask emerging adults to discuss their experiences in the present but also to reflect on their experiences with identity and influences on identity during childhood so that we can retrospectively understand the changing influences throughout their life to this point. An intersectional framework is critical because there are other aspects of identification including gender, religion, geography, and socioeconomic status that influence how a person identifies and influences how they are perceived by others.

The framework presented demonstrates the multiple spheres of influence that are necessary to account for when exploring how an individual identifies and is identified. No two individuals will have precisely the same influences on their racial and ethnic identity development. Multiracial individuals should not be looked at as a monolith, but instead as individuals who have different intersecting experiences and influences that lead to different identity developments and also potentially different outcomes. Previous research has demonstrated that multiracial individuals are often racialized to be included in a monolithic racial group. This means that multiracial individuals may internally identify a certain way but they are treated differently based on external characteristics or assumptions by other people. This discordance in personal identification and how the world treats an individual can be very stressful and create psychological conflict [8]. It is therefore critical to understand both how individuals self-identify but also how they are identified by others and what they do if those two identities do not match.

Although this framework includes components about the influence of identity development on stress and health, that will not be the focus of this paper. This analysis instead focuses on multiracial identity development throughout childhood and emerging adulthood and the many intersecting influences on that development.

## Study aims

In the context of emerging adulthood and early adolescence, self-identification is a process that may shift as the following variables are introduced: changes in geography, socialization

education settings, and parental and family influence. The self-identification distinctions multiracial adolescents and young adults adopt during these changes are important to capture. Given that these changes are difficult to categorize, it is imperative that researchers continue to research the variables affecting multiracial identity formation during adolescence and emerging adulthood because this life stage has critical implications for the socioemotional and physical health of individuals. This study uses qualitative interviews with emerging adults to explore individual, interpersonal, and structural influences on identification. We ask participants to reflect on experiences from childhood through their present time as emerging adults.

## Methods

This study was a qualitative study of emerging adults (ages 18–29) that identify as multiracial. This study was approved by the CUIMC Institutional Review Board. Participants were walked through an information sheet at the beginning of the zoom interview and were able to ask questions. After questions were answered, the participants were asked to provide verbal consent (written consent was waved so that we did not have names of participants attached to interview transcripts).

### Setting and participants

In-depth qualitative interviews were conducted with 21 participants from across the United States. Participants were recruited from multiple social media platforms including Instagram and Facebook on pages specific to multiracial young people. A virtual flyer was posted in these groups asking self-identified multiracial individuals between the ages of 18 and 29 to respond if interested in taking part in a qualitative interview about their experiences about identity and identification throughout their life. Due to the recruitment sample, our participants are skewed toward those in higher education programs.

### Data collection

Interested individuals responded to social media recruitment and emailed the study team if interested in being included. The study team then reached out to set up a zoom interview with the participant.

A semi-structured interview guide was developed by the study team using a conceptual framework theorized by Grilo et al. (2022) and through reviewing recent literature on multiracial identification. The semi-structured interview guide was adjusted as we learned from participants and identified questions that were helpful or not helpful to include in the interviews. The interview guides included eight sections that focused on the following topics: racial and ethnic identification, background and growing up, parental and family influence, peer engagement and treatment, health and wellbeing, experiences of discrimination, forming resilience and language and terms. Although the interviews all took place with emerging adults (ages 18–29), we asked participants throughout the interview to discuss their experiences from childhood through to the present time to understand how identity development and their multiracial identity has been influenced throughout their life course. A brief series of two demographic questions were also asked, age and level of education, and a concluding question asking if there was anything the participant wanted to add that wasn't discussed was used to close the interviews. The full interview guide can be accessed by request of authors.

Semi-structured interviews were conducted with all participants by two research assistants. Interviewers were trained in qualitative research and how to properly use semi-structured interview guides. They followed the structure of the interview but also used probes and prompts where necessary to elicit further information and were given leeway to adapt or

expand certain areas of the interview guide based on the conversation and information the interviewee wanted to share. The interviews lasted about one hour each and were conducted on HIPAA compliant zoom. Participants received a $50 dollar Amazon gift card which was emailed to them after their participation. The audio from zoom was recorded and transcribed for analysis and then uploaded to Dedoose for coding and analysis.

## Data analysis

The research study team used a thematic framework analysis method to analyze the data [9]. The conceptual framework was utilized to develop the interview guide in this qualitative analysis, as well as to understand and organize the data display and results. The researchers each independently reviewed transcripts to generate initial codes. A preliminary codebook was developed based on the initial code generation. This codebook was reviewed and tested by the two primary coders and revised until no new codes were identified. All disagreements were discussed, and the main coders went back to the codebook to clarify definitions more clearly before finalizing the coding of the remaining transcripts. A third member of the research team reviewed the final codebook and the preliminary coding of the first two transcripts. The final codebook was applied to all remaining transcripts. After coding was complete, a data display was created to analyze the data. The data display was organized to explore the discussion of different factors (i.e., geographic location, childhood and upbringing, perception by others, etc.) in relation to racial and ethnic identification, self-identification, and other salient identities.

## Demographic findings

*Table 1* displays the demographic characteristics, including age, pronouns, education status, and geographic region of the 21 participants.

The majority of participants fell within the 25–29 age group (61.9%) and the majority of participants used she/her pronouns (85.7%); however, it is notable that nearly 10% of participants used both she/her and they/them pronouns (9.5%); additionally, less than 5% of participants used they/them pronouns alone (4.7%). Nearly half of participants (47.6%) were in the process of earning a degree, with the majority of those participants earning a doctoral degree (*bachelor's* 4.7%; *master's* 14.3%; *doctoral* 28.6%). The remaining participants (47.6%) had completed a degree at the time of the interview and did not disclose whether they were in the process of earning an additional degree (*bachelor's* 23.9%; *master's* 19.5%; *doctoral* 4.7%).

The last point of demographic data collected, outside of multiracial identity, was geographic region residency. Participants were asked about where they grew up–which for most, included many states, cities, and even countries–and where they were living at the time of the interview. All participants were currently living within the United States. Participants' current geographic residency was categorized via seven United States' regions: Northeast, Southeast, Midwest, Pacific, Rocky Mountains, Southwest, and Noncontiguous. The majority of participants were currently living in the Midwest (38.1%) and Southeast (28.6%) regions. The Northeast (14.3%) and Pacific (9.5%) regions were moderately represented at the time of interviews; with the Rocky Mountains (4.8%) and Southwest (4.8%) regions being underrepresented. No participants were residing in the Noncontiguous region at the time of the interview.

## Thematic findings

**Grilo et al.'s (2022) theoretical framework was** in the formation of an organizational and comprehension scaffolding for results analysis. This framework explores how factors on the

**Table 1. Sample characteristics for multiracial identification and health: Belonging study.**

| Multiracial Study Respondents (N = 21) | | |
|---|---|---|
| **Key Variables** | **N** | **%** |
| **Current Age (years)** | | |
| 18–24 | 6 | 28.6% |
| 25–29 | 13 | 61.9% |
| 30 | 1 | 4.7% |
| Missing | 1 | 4.7% |
| **Pronouns** | | |
| She/Her | 18 | 85.7% |
| They/Them | 1 | 4.7% |
| Both | 2 | 9.5% |
| **Education Status** | | |
| Pursuing BA | 1 | 4.7% |
| Completed BA | 5 | 23.9% |
| Pursuing MA | 3 | 14.3% |
| Completed MA | 4 | 19.5% |
| Pursuing PhD | 6 | 28.6% |
| Completed PhD | 1 | 4.7% |
| Missing | 1 | 4.7% |
| **Geographic Region** | | |
| Northeast | 3 | 14.3% |
| Southeast | 6 | 28.6% |
| Midwest | 8 | 38.1% |
| Pacific | 2 | 9.5% |
| Rocky Mountains | 1 | 4.8% |
| Southwest | 1 | 4.8% |
| Noncontiguous | - | - |

individual, interpersonal, community, and structural level influence one's identity development over the life course. Critically, this framework emphasizes that these factors may influence identity formation differently depending on what stage of life an individual is in. Therefore, this results section is organized to explore major influences during childhood and then during adolescence and emerging adulthood.

Interviewees were asked about individual and interpersonal influences on the formation and expression of their racial and ethnic identity. Identities that were reflected upon–whether from the old, new, or evolving–were described by interviewees in three different, but interconnected, facets: self-identification, racial and ethnic identification, and other salient identification. Self-identification was defined as "how a person describes their physical features, or phenotypic characteristics".

> I think my hair has something to do with it. I tend to see more biracial kids that come from black and white. Both have a certain type of curl texture, and it's a lot thicker than mine, but it still has that curliness to it. I got my mom's hair, straight hair in texture, but the curl pattern is my dad. They'll automatically be like, oh, you must get the nice hair from another race, other than white or black, whatever. Also, my skin, I live in <City>, and I'm always in the sun, so they're just oh, the skin tone and all that. (MIH 11, 27, Pacific)

Racial and ethnic identification was defined as "how a person chooses to categorize themselves in a racial or ethnic group based on a sense of belonging, shared culture, history and/or shared ancestry".

Interviewer: How do you identify in terms of your racial, ethnic background?

Interviewee: I identify as biracial, multiracial woman.

Interviewer: Could you go in a little more in depth into what your racial ethnic background is?

Interviewee: Yes. I will say yes, I would definitely go with multiracial, but my backgrounds are native, white and black. (MIH 04, 23, Northeast)

Other salient identifications were defined as "how a participant discusses other important or notable identities that are race or ethnicity, such as socioeconomic status, immigration status, gender identity, educational status, etc.".

I would think the one thing that defines me most, just with my background and being in a lot of spaces and something that I find commonality across cultures and genders, and everything is just being low income. Coming from a low-income background, financial concerns are, I mean, financial concerns, race concerns, gender concerns, all these are at the forefront of every day of my lived experience every day. I would say it's the forefront struggles that have maybe caused me the most harm or difficulties or challenges. How am I going to eat? How am I going to pay rent? How am I going to be able to go to college? How am I gonna be able to make it through the semester? I would say being low income has been something that's just been so hard for me even to get into any of these spaces. (MIH 20, 26, Midwest)

These three facets of identity were formed and expressed by internal and external influences in interviewees' lives, such as their childhood and school experiences and family and peer relationships.

## Life course

It became clear through talking to participants that the different influences on identity (individual, interpersonal, community and structural level factors) influenced their multiracial identity development differently at different stages of their life. Thematic results are therefore discussed in two separate life stages: childhood and emerging adulthood.

**Childhood and identity.** Participants were asked to describe their childhood experiences, which included the geographical and demographic details of where they grew up, their family history and kinship network, and school and peer experiences. The results for the childhood portion of the life course fall into two main themes: identity discovery and identity expression and reception.

**Identity discovery.** Childhood proved to be a time of self-discovery and identity formation. Participants' experiences, rearing and parental influences, conversations and interactions with peers and family all served as factors in participants' identity discoveries. The narratives that participants are told growing up surrounding their racial and ethnic ancestry can lead to an understanding of identity that may not hold true in adulthood, and may take years to unlearn and rediscover:

Well, from early childhood we're told in Puerto Rico that we're a mix of African, of European and of Taino, so it's an understanding across the board. Even though there are white

Puerto Ricans, so it's a problematic narrative. There are white Puerto Ricans, there are Black Puerto Ricans, and there are Puerto Ricans who look mainly Indigenous, but there's always been the narrative that we're all mixed. That's not necessarily true. For me it's always been that way, but only in recent years was when I started learning about how there's differences in our identities. There's differences in our heritage, even though we're from the same island. (MIH 15, 30+, South)

The information that is given to children by their parents and family about who they are and the position they hold in the surrounding world has lasting impressions and influence on their identity–especially for individuals of mixed-race backgrounds. A factor that most often presented as an influence during childhood identity development and parental influence was participants' physical features, especially "light skin" or white presenting. Participants discussed differences in siblings' physical features, or children's physical features from their parents and family.

Yeah. I've talked with them about this, too. They always say that they identify as Mexican, Mexican and white, and that's usually what they tell people, too. Then it's interesting because me and one of my sisters are very light skinned. Then I have one brother and one sister who are very dark skinned. When you put us together, a lotta people actually say, 'I didn't know that was your full brother. I didn't know that was your real brother.' I was like, 'No, we're [laughter] really brother and sister'. (MIH 20, 25–29, Midwest)

These differences in physical features amplify perceived differences from surrounding family's identity, culture, and belonging. Perceptions from others regarding participant identity are not limited to strangers on the street, or non-family members. In fact, an amplification of belonging and identity exists within kinship networks for multiracial youth as they navigate their identity and the identities of their surrounding families.

I identify as a Mexican American because my dad has—he's half White and my mom is fully Mexican. I've always been more around my Mexican family, but obviously there's still that White identity, a portion of me that I feel like I don't really connect with. It makes me feel different with my Mexican family because it's like they look at me different. Like, "Oh wait, you're not fully Mexican though or as traditional," or stuff like that. (MIH 12, 25–29, Pacific)

Different racial and ethnic backgrounds pose unique challenges for multiracial youth beyond kinship relations and acceptance. As children age, they begin to rely more and more on the relationships and complexities of their social lives, especially amongst peers. Where participants grew up, who they were exposed to and had opportunity to learn from and connect to during their formative years plays largely in identity formation and community attachment.

In <City>, my family's a lower middle class, and we've always been since I was born. In <City> we're always around other kids that were in the same class majority, and the population is just mostly African American. . .I went to a school that was very diverse in every single race. It was definitely a changeup because even though I could still relate to the Black community because I'm Black, I didn't have many multiracial friends to relate to or understand. When I came here, I did have a lot more people I can relate to that have different backgrounds, but also can understand being more than one race in <City>. (MIH 11, 18–24, Pacific)

Exposure to certain communities during childhood can lead to misunderstanding of identity like the ways in which family narratives can lead to unclear or incomplete identity formation. The demographic makeup of the community mixed race children are mired in is another prominent avenue identity exploration and formation occurs. Stumbling upon their identity through misunderstanding based on the surrounding demographic, was a theme for some participants who grew up in diverse areas and were exposed to varying identities–a recipe for identity discovery, experimentation, and provoking communication with family.

> Yeah, well, discovering my whole racial identity has been a really long process. I grew up in <State>, I was born and raised there, and most people in <State> are mixed race, and you can't tell really someone's ethnic or racial background just based on how they look, 'cause everyone's mixed, but because I grew up in <State>, I thought I was Hawaiian, and I didn't realize—well first of all, I learned in third grade that I was white, even though my mom is white, I did not know that I was also white until I got teased on the playground, and then I talked to my mom. They used a kind of derogatory term for white people, that is common in <State>, and I didn't really know what that meant, but I asked my mom about it, and she was like, "Yeah girl, you're half white," and I was like, "What? That's so crazy,". (MIH 06, 18–24, Midwest)

Participants reflected on understanding their identities as a trial-and-error string of events–testing out identities to see how they fit not only their own understanding of who they are and who they come from, but also their family's own identity and understanding of race and ethnicity. This trial-and-error type of discovery and identification is further influenced by familial correction and amending.

> [A]s I've been—growing up in high school and college, I've embraced biracial as a term, and I think I did that to—for myself to not just be—not to think of myself as half of two different cultures, but to be a whole person that is made up of different cultures, so I went with biracial, but I think in the last year or two, I've been moving away from biracial and toward mixed race, because I'm pushing back against this notion—again, in a different way about —that I am half of something and half of another thing, because I feel pretty mixed, so, I use mixed race now. (MIH 06, 18–24, Midwest)

Communication, whether with family or community members, such as schoolteachers or peers, is a pathway for identity early on, where physical differences within and outside of families–often an integral facet of childhood understanding, and the narrative children exist within and view the world through–can be further explored and understood.

Yeah, I mean, I think it was pretty early on. I can very easily remember talking about it in the first grade, so as early as that, but likely earlier. I mean, my mom looks very different from how my sister and I look, so I think that pretty early on, it was evident to us, but it was a huge part of our lives. (MIH 07, 25–29, Northeast)

Throughout these examples of childhood self-discovery and identification is the concept and event of communicating: communicating with family, friends, institutions, and community members. Communication, especially early on, proved to be an essential factor in understanding identity–both self and familial–for participants. The language that is used strategically—but often coincidentally–proves to be elemental in identity formation, understanding, and expression. The influence language has within family structures, as seen with this participant, goes to show just how nuanced identity can be.

My sister is complicated. I think I identify more closely with the Latino heritage than my sister, who identifies more closely with the Irish heritageI don't know, sometimes my sister—she identifies as Latino to a point, I guess, just to a lesser extent that sometimes I'm worried that I'm faking it, or that my connection to the culture is entirely made up by me and it doesn't actually exist. (MIH 05, 25–29 Northeast)

**Identity expression & reception.** Participants' identity actualization started with childhood experiences and interactions. With a newfound or discovered identity, participants began expressing these identities in spaces and at times, reacting to and restructuring their identities based on the reception their expression received and the perceptions from those around them. Participants discussed how their phenotype and the way their identity is expressed often changes how they are received, and that this treatment is sometimes different from treatment their families receive.

I think it totally influences it. I have red hair. You don't usually get people with red hair who are mixed race. . .Obviously, I benefit from a lot of privilege like that. Sometimes if my father asks me to come in a store with him, because if he goes alone the people are going to mistreat him. If I'm there, they treat us both better. I think it's given me a lot of social opportunities, just because of colorism. Obviously in Latin America, that's not really a Latin American race, and we always forget at the afro Latinos and whatever, colorism is a huge thing. (MIH 04, 18–24, Northeast)

Participants also discussed how regardless of how they choose to identify, how others treat them also impacts their identity and their feelings of belonging.

I identify as a Mexican American because my dad has—he's half White and my mom is fully Mexican. I've always been more around my Mexican family, but obviously there's still that White identity, a portion of me that I feel like I don't really connect with. It makes me feel different with my Mexican family because it's like they look at me different. Like, "Oh wait, you're not fully Mexican though or as traditional," or stuff like that. (MIH 12, 25–29, Midwest)

Other individuals expressed a unique experience–that it was not that they came to a multiracial identity later, but instead that they and their family always knew and embraced being multiracial.

I would say I always knew that my mom and dad were different. I always knew that he's Mexican, she's Irish now Scottish. That was always just two parts of my identity. I thought it was interesting because I had the best of both worlds. I got to have Southern cooking from my mom's side, and then I got to have really great Mexican food from my dad's side. It'd just be the best of both worlds mixing together. There was never a point where I was just like—I knew that I was multiracial. I knew that I wasn't 100 percent either, I would say. I knew that pretty early. I was like, "I know that I'm not 100 percent my dad. I'm know I'm not 100 percent my mom. I'm a mix of both (MIH 20, 25–29, Midwest)

## Emerging adulthood

The emerging adulthood stage in the life course is a critical time in identity formation as participants experience how the perceptions, attitudes, and behaviors of people outside of their family and childhood community influence how they identify ethnically, racially, and at times, both. This section is organized into four central themes: the impact of changing geography, nurturing connections with peers by creating a shared sense of community, strategic disclosure, and continuity of racial and ethnic identity formation.

**Changing geography.**    Participants shared the influence of changes in their environment on their identity formation, specifically with changing geography as a catalyst for shifts in their self-perception and treatment of identity. As they moved from their childhood communities to college, the demographics of their environment oftentimes shifted drastically. As a result, participants discussed how their salient identities shifted along with their physical migration.

One participant, living in the Midwest shared:

Then I moved to <City>, Iowa, which is a red, racist state. They're stuck 50 years in the past, and being Black became again my salient identity because I wore it. It's what people see me as. Being Black and being female. As I was working through my trauma and racial identity and stuff, I came into my trans identity. (MIH 01, 25–29, Midwest)

The same participant shared how their multiracial identity shifts along with the demographic makeup on their environment–that in settings where there were few people of color, they were seen as Black and identified as Black. When they were in spaces or environments with more people of color and felt able to have a more nuanced identity, they identified as multiracial. The concept of the one-drop rule was also discussed by this participant. Any indication or perception by others of being non-white, they shared, signaled an "other" or "non-white" designation:

I went from experiencing seven percent of Black people to less than one percent of Black people, and so that was shocking. The smaller it got, the more Black I got [laughter]. You have any kind of melanin, you're Black, or you're not White. In Iowa, I identified as Black in all spaces, and then I would say in college and being in <Region> for my adult life, I identified more as biracial. (MIH 01, 25–29, Midwest)

The intersection of the racial/ethnic makeup of an environment with how an individual appears or is perceived, makes for different experiences for different individuals. The participant above identified as Black because they were perceived as Black in a place with very few people of color. For the respondent below, who was often perceived as White, the homogenous environment they found themselves in led them to identify as White rather than multiracial.

Because I've always been surrounded by White or White-passing individuals, it was no stretch for me to identify as White, even though ethnically and culturally, I wasn't White when I first moved here. I'm more so White, now. I married a guy from small-town America, and that always makes for a fun conversation. (MIH 08, Northeast)

**Community discovery.**    Most of the study participants had a common experience of using college and graduate school as spaces to gather and develop a vocabulary for identification.

After I went to university, I finally got the vocabulary to talk about these things and learn more about the impact of colonialism in Latin America, I have a more in-depth understanding of these broader issues. I think I've always identified as not white. That hasn't changed, but the nuances within that have changed as I've gotten older, particularly, within the past 10 years. (MIH 03, 25–29, Southeast)

Participants also discussed that it was during this period of emerging adulthood that they began to think more critically about identity, race, and began to make meaning of experiences they had already had.

When I was younger, I wasn't talking about race. I didn't feel like I was looked at differently because of my race just because I was still around like a majority of African Americans. Within the past three years, I've gone through an identity crisis, and especially with Black Live Matter, I've educated myself. When I was younger, I didn't really know what to identify as. People would force me to put me into one or the other when I can really just identify as both. Being in <City> it's totally different. (MIH 11, 18–24, Pacific)

Another participant shared:

As I got older, and especially as I started thinking more critically about race and other constructs of identification in college, I really started realizing I was treated differently at times, or people expected me to be extremely smart because I was Asian. Even though to me that model minority myth is mostly associated with East Asian people, I think that it was mainly because there were so few Asians in general in town, but that sort of got attached to me as well. (MIH 07, 25–29, Northeast)

Participants also brought up how the racial make-up of their school or community influences how they identify and sometimes, how that identity shifts over time. Participants shared the impact of the racial and socioeconomic demographics of their educational settings on their self-identification. They also shared how this change over time influenced their identification:

My college was a private, religious university, and it was 96 percent White, so identifying as White there was just a result of fitting in. In college, I majored in Sociology, then I got my Masters, and I'm getting my PhD in Sociology. It's been a coming to terms with multiracial identity; kind of accepting that minority identity and balancing the two, because I still identify as White, racially. (MIH 07, 25–29, Northeast)

They also discussed how when in homogenous environments it is often difficult to feel like they have a community or fit in. One participant described the barriers to forging close friendships with peers. She expressed feelings of uneasiness when invited to socialize with a monoracial group of classmates:

It been hard to build close relationships with the people in school. Sometimes they'll ask me to hang out, but it's kind of hard for me to want to. I hate the idea of being out with a group of 12 white women and I'm the only Black person there. That's very uncomforting to me. (MIH 21, 18–24, Southeast)

**Strategic disclosure.** Many participants discussed the idea that in their past they identified as White (and not as multiracial or other identities) to feel as though they fit into their community of predominantly white peers. Experiences of racial isolation and not having peers or a community compounded these experiences. One participant who perceived this form of privilege shared her experience:

In high school, I lived in a predominately White area, pretty socio-economically advantaged, and so for me to make the most of my experience socially and educationally, identifying as White was really the most beneficial. It provided the most advantageous social circles as well. That was a time when I felt White. So, it's not that I only identified as White, I also felt White as well. (MIH 08, Northeast)

Another participant explained that they used to really protect their Mexican identity and not share it widely, but that they came to realize how important that Mexican identity was to them and that even though phenotypically they were often perceived as White, being Mexican was a large part of their identity and they wanted to share that with others. Participants shared the protective and sometimes violent implications of not sharing their non-white identity with others. When first interacting with others in majority-white settings, one participant shared, volunteering information about her Mexican heritage was not comfortable.

I used to try to avoid telling people that I was Mexican at first because a lot of people assumed that I was white. It was something that would really try to be protective in predominantly white spaces. Later, I would feel more comfortable sharing my full ethnicity and full identity and telling people that, yeah, I am Mexican, and my father is from Mexico. Even though I am white-passing or am much lighter-skinned, that is still a salient part of my identity. (MIH 20, 25–29, Midwest)

This same participant mentioned that because she was "white-passing" sometimes people would say upsetting things around her not knowing her Mexican identity, which also influenced her to disclose her full identity to them. She went on to share an experience of being subject to racist comments by a group of people who believed they were in an all-white space.

I know that people only feel comfortable saying those things because they assume that I'm white. It's protective sometimes to not always proclaim your identity. Sometimes it gives me privileges, but then I might find myself in a situation where I'm around people who are saying things that are really upsetting to me. Then, I have to come out and disclose and say, "This is who I am. I find this really problematic." People feel comfortable saying those things because they assume that I'm white at first. They would have never said that if they knew I was Mexican. (MIH 20, 25–29, Midwest)

Some participants expressed feelings of guilt or betrayal toward one side of their identity that they may not have always been identifying with. Participants expressed a process of self-education about racial identity in their formative years. Equipped with this knowledge, they would express a desire to claim parts of their racial identity other than whiteness. The concept of whiteness as default or neutral identity results in feelings of betrayal of their non-white identity. One participant expressed:

Being surrounded by White people at every point in my life, even in different countries, made it easier for me to identify as White. As I got older and learned more about racial identity and the oppression that comes with it, it—having been surrounded by so many White people also made me want to identify as Hispanic. I felt like, in a way, I was betraying that little side of me. (MIH 08, Northeast)

Some participants also brought up being hesitant to identify a certain way around certain groups, for example as a light skinned person being uncomfortable identifying as multiracial or Black around Black people for fear that others would not accept that or be upset by it. Participants expressed feelings of hesitancy in their youth in sharing their self-identification, particularly when their skin tone afforded them proximity to whiteness. The roots of this hesitancy they shared were often concerns about how monoracial people in interpersonal settings might react to their self-identification.

When I was younger, I was more hesitant to refer to myself as Black around Black people because of my light skin privilege, and I didn't want to step on any toes or feel like I was conflating my experience to their experience. I grew up White, in a predominately White area, so I didn't "feel Black." In college and post-college, I was adamant about identifying as Black around White and Black people. There's still some hesitancy there. Now I'm 27, so now I'm pretty firm in my identities so no matter the space I am in, I identify as Black. If there's more to say, I'll say a light-skinned Black or biracial Black person. (MIH 01, 25–29, Midwest)

**Identity formation continuity.** One of the key concepts that came up throughout the interviews was that identity formation is not a process that happens linearly or happens at one time and is then complete and static. Instead, different participants shift their understanding of their identity over time, shift the terms they use over time, and for many this process is still evolving. One participant mentioned that they have come to understand what biracial means over time–not that they are halves but that they are made up of multiple cultures. A key finding shared by multiple participants is the concept of identity formation as an evolving, non-linear process. One participant shared that they have redefined what the term "biracial" means to them as they have developed over time. Their new self-definition states that rather than being halves, they are one, fully-formed person with a rich and multifaceted set of cultures:

It took me a while to get those facts about myself. Growing up in high school and college. In the last year or two, I've embraced biracial as a term, and I did that to not to think of myself as half of two different cultures, but to be a whole person that is made up of different cultures. (MIH 06, 18–24, Midwest)

Other participants discussed how because they appear White to others and can pass as white, they feel they cannot identify as a person of color or join those communities because of the privilege they have through their whiteness. Other participants shared how their phenotypical presentation as white-passing influences the social groups in which they choose to engage. This decision is in part influenced by an acknowledgment of the societal privileges that being read as white passing provides. One participant shared:

When there's groups for people of color, I tend to not opt into those groups, because I benefit from white privilege, and my dad is white, so that makes me white. Even though I don't feel like I fully fit into whiteness, how do I use the labels that we have with our language and with our country here? (MIH 18, 25–29, Southeast)

The process of identity formation is often uncomfortable and often does not feel "complete" but as something that individuals are navigating throughout their life. The importance of acknowledging privilege was discussed by most participants (N = 16). Self-identification as a non-linear process was described by one participant:

Saying that I'm a white Latina doesn't feel accurate, but then not acknowledging my whiteness, when it's such a privilege, especially within a Latinx community also doesn't feel accurate. My identity hasn't been solidified. It's still something that I'm trying to figure out what feels most authentic to me, but also what feels an accurate acknowledgment. (MIH 10, 25–29, Southeast)

## Discussion

The current research delved into the qualitative data of multiracial emerging adults whose experiences of forming and maintaining a multiracial identity intersects with the multiple

complex facets of the socioecological system. These intersections and resulting experiences vary depending on the positionality of the participant within the life course, with identity and socioecological facets interacting with differing frequency and intensity. Participants, reflecting on early life experiences–*childhood*–and later in life experiences–*emerging adulthood*–illustrate the varying degrees in which their multiracial identity interacts with their other identities, relationships, and contexts, and how this informs the identity formation process.

There were three identified facets of identity that participants discussed in relation to their life course experiences with their identity formation, expression, and interaction with the socio-ecological environment. These facets of identity were their racial and ethnic identity, their self-identification, and their other salient identities. These three identity facets were involved in much of the intersectionality of the identity formation process; proving to have varying degrees of interactions and influence depending on where in the life course individuals found themselves and with what factors of the socio-ecological environment they were interacting with.

## Childhood

In the beginning of the life course, *childhood*, participants reflected on how this was where many, if not most, discovered what identity meant to them and encompassed. *Identity discovery* mainly occurred for participants regarding their multiracial identity, however their self-identification or phenotypic characteristics and other salient identities played a significant role in their identity discovery. Interactions within the context of the socio-ecological model, specifically social interactions, were influential on participants' identity discovery. Such influential interactions included those with parents and family, as well as peers, community members, and classmates. Specifically, the conversations that participants had during childhood revealed much about their own identity, as well as the world around them. Differences–whether perceived or informed by others–influenced participants' identity formation and subsequent interactions within the socio-ecological model. These interactions had consequential significance on participants' sense of belonging: to family, to community, and to an identity. As participants traversed the life course and began to rely more and more on peer interactions and regulation, the identity formation process changed, with novel weight being applied to the opinions and interactions with peers and communities. These new interactions with garnered importance prove to be difficult to navigate, especially for multiracial youth who maintain differential racial and ethnic backgrounds. These trial-and-error relationships, community memberships, and identities are due in part to misidentification or misunderstanding from others and from oneself, but also is an elemental aspect of the identity formation process–especially early in the process, during childhood. Throughout the process, experiences, and changing reliance on key persons, participants consistently reflected on the event and concept of communication. Communication, whether with family, friends, institutions, or community members, was indispensable to identity formation and intrinsic of the process.

The communication and interactions participants had throughout childhood with their surrounding social and ecological surroundings led their identity formation process to actualization. Through expressing their identity and integrating the various responses and receptions from those around them. As they expressed their identities in spaces and at times, their phenotype, or self-identification, and their identity expression interacted with those around them, leading to changes in their identification. Regardless of how participants choose to express their identities, there are challenges with others' perception of their identity. However, for some participants, strong family and parental narratives and understandings regarding their multiracial identity enabled participants to understand their identity from an early age, overriding much of the identity formation facets and occurrences that other participants expressed.

## Emerging adulthood

Data from this study reveals there is an interaction between geography and identity on the experiences of participants. Participants describe the demographic characteristics of the areas in which they spent their childhood as compared to the demographics of the towns and cities in which they relocated for college or university programs. Most participants we spoke with currently live in different states from which they developed their early childhood and early adolescent understandings of their multiracial self-identification. The experience of physically relocating oftentimes resulted in a shift in perspective regarding in which categories they felt they belonged demographically. For example, several participants shared the experience of becoming whiter or becoming less "other" once they moved from their homogenous childhood geographic regions.

Participants shared how their external identity formation, or the process by which other people influence their perceptions, attitudes and behaviors on how identify ethnically or racially has changed over time. The influence of their chosen communities, most of which consisted of their college or graduate school peer groups, with whom they share common characteristics, was one significant factor in the racial/ethnic identity of participants.

The findings regarding at which points in their life course participants have felt protective or uncomfortable sharing their racial or ethnic identity in social settings underscore an important finding in the existing social science research on the effects of colorism. Previous research has shown for example, that Mexican Americans of lighter skin tones fare better economically when compared to medium-tone Mexican Americans [10]. Previous models of multiracial identification demonstrate the critical influence of structural factors which contribute to the formation of multiracial self-identification which can't be overshadowed by individual or community factors alone [3].

All participants also discussed other salient identities including gender identity, socioeconomic status, class and immigration and their impacts on their understanding of their identities. This research contributes to existing knowledge by seeking to understand what factors influence racial and ethnic self-identification as multiracial adolescents emerge from the childhood life stage to early adulthood.

## Limitations

Due to the sampling method used, in which participants were recruited from social media groups for higher education students, a limited sample was gathered. The social media groups targeted for sampling limited the sample to individuals in higher education programs (bachelor's to doctoral students), with the majority having completed a master's and pursuing a PhD. This sampling bias in terms of overrepresentation in higher education could have also contributed to the fact that we had a majority of female-identifying individuals.

Underrepresentation also existed within the participant sample regarding geographical location of participants at the time of the study. Participants were underrepresented in the Southwest and Rocky Mountains regions, with no participants currently living in Noncontiguous–Alaska and Hawaii–region at the time of the interviews.

Participants were asked to disclose their pronouns at the outset of the interviews to facilitate peer camaraderie between participants and interviewers, as well as to discern participants pronoun identification. The researchers chose to ask participants their pronoun use rather than gender because pronouns were deemed more indicative of their identity formation and expression, as they can, for some, represent their gender identity expression and are used in social interactions. However, it is recognized by the researchers that pronouns should not and are not conflated with gender identity. There are limitations associated with asking participants

their pronouns. This data is not generally representative of participants' gender or sexuality identity, both of which were discerned through *some* participants disclosure of *other salient identities*. However, it is regarded as missing data about the demographic details of participants regarding gender and sexuality information.

Using a top-down approach or a pre-existing framework to understand qualitative data can also be considered a limitation. Although a framework analysis is helpful in that it situates the data within a larger context, using a pre-existing framework does also introduce some level of bias in how the researchers approach coding and analyzing the data. The research team tried to mitigate that bias by having frequent meetings when discussing the creation of the codebook and the coding process and in the data display creation and analysis to ensure that the final themes were true to the data and participant experience.

## Implications for future research

The current understanding of what the United States' multiracial population consists of has grown, and will continue to grow, as reporting and measuring tools improve. The drastic increase in the Census multiracial population measure between 2010 and 2020 is a clear indication of this. However, the importance of understanding and recognizing this growing population does not end with the size of the population. The formation, expression, and reception of identity in general, especially a multiracial identity, is a process in which little has been studied and recorded. Multiracial individuals are numerous, but the ways in which their identities form have hardly been studied [11]. Overall, an area of research that is convoluted and homogenous–not representative of the vastness that is identity and lived experiences [12, 13].

Grilo et al. [3] has begun the expansion of the relevant study of identify formation, theorizing the framework in which the formation process can best be understood. The framework, Ecological Framework for Understanding Multiracial Identity Development, posits that not only is the ecological fabric in which individuals are mired in influential, but that the life course itself proves to be significant in the identity formation process. The current qualitative research illustrates the ways in which the intricate network of ecological layers interacts with individuals' life course, resulting in unique and poignant experiences and identities [14–16].

Future research investigating the ways in which identity, specifically multiracial identity, is formed and expressed should be framed within the *Ecological Framework*, emphasizing the intersectionality of the ecological environment and factors in different capacities along the life course [4]. Additionally, future research should look to expanding the sample of anecdotal and narrative evidence beyond the characteristics of those who participated in the study. Individuals from all regions of the United States, as well as internationally, should be recognized and included in such research as to further understand the geographic influences on identity formation and expression, as well as interactions with other socio-ecological factors. Life course events and phases may differ for individuals outside of the United States, considering the differences in cultural practices, child-rearing and schooling to name a few. It is also critical that research examines the health effects and outcomes related to this identity development process for multiracial individuals [16].

## Author Contributions

**Conceptualization:** Stephanie A. Grilo.

**Data curation:** Stephanie A. Grilo, Mollie R. Semler, Stephanie Rameau.

**Formal analysis:** Stephanie A. Grilo, Mollie R. Semler, Stephanie Rameau.

**Funding acquisition:** Stephanie A. Grilo.

**Investigation:** Stephanie A. Grilo.

**Methodology:** Stephanie A. Grilo, Mollie R. Semler, Stephanie Rameau.

**Project administration:** Stephanie A. Grilo.

**Resources:** Stephanie A. Grilo.

**Software:** Stephanie A. Grilo.

**Supervision:** Stephanie A. Grilo.

**Validation:** Stephanie A. Grilo.

**Visualization:** Stephanie A. Grilo.

**Writing – original draft:** Stephanie A. Grilo, Mollie R. Semler, Stephanie Rameau.

**Writing – review & editing:** Stephanie A. Grilo, Mollie R. Semler, Stephanie Rameau.

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
