## [Decision Letter · Decision Letter 0]

20 Oct 2022

PONE-D-22-12004The sum of all parts: A multi-level exploration of racial and ethnic identity formation during emerging adulthoodPLOS ONE

Dear Dr. Grilo,

Thank you for submitting your manuscript to PLOS ONE. After careful consideration, we feel that it has merit but does not fully meet PLOS ONE’s publication criteria as it currently stands. Therefore, we invite you to submit a revised version of the manuscript that addresses the points raised during the review process.

We look forward to receiving your revised manuscript.

Kind regards,

Pieter-Paul Verhaeghe

Academic Editor

PLOS ONE

Journal Requirements:

3. We noted in your submission details that a portion of your manuscript may have been presented or published elsewhere. [Figure 1 is the conceptual framework first created by Grilo et al. and used as the grounding framework for this study (2022)] Please clarify whether this [conference proceeding or publication] was peer-reviewed and formally published. If this work was previously peer-reviewed and published, in the cover letter please provide the reason that this work does not constitute dual publication and should be included in the current manuscript.

4. Please include a caption for figure 1.

5. Please include your tables as part of your main manuscript and remove the individual files. Please note that supplementary tables (should remain/ be uploaded) as separate "supporting information" files

Reviewers' comments:

Reviewer's Responses to Questions

**Comments to the Author**

1. Is the manuscript technically sound, and do the data support the conclusions?

Reviewer #1: Partly

Reviewer #2: Yes

2. Has the statistical analysis been performed appropriately and rigorously? 

Reviewer #1: N/A

Reviewer #2: N/A

3. Have the authors made all data underlying the findings in their manuscript fully available?

Reviewer #1: Yes

Reviewer #2: Yes

4. Is the manuscript presented in an intelligible fashion and written in standard English?

Reviewer #1: Yes

Reviewer #2: Yes

5. Review Comments to the Author

Reviewer #1: I think that this dataset represents a unique source of information, and the theoretical approach is a strong contribution to understanding multiracial identity development from multiple angles. My primary concerns are a lack of clarity around how health and life course factors fit into the interview approach and data. Although some of this may be fixed with revision or perhaps selecting quotes that are more representative of the data, it can also be indicative of the researchers imposing their own framework on data that doesn’t adequately reflect all parts of this framework (e.g., lack of health discussion in the results and yet is discussed as a major contribution of the paper). More methodological detail would assuage reader concerns regarding the latter. Below I outline my primary comments. I hope the authors find this feedback helpful.

1. Although the conceptual framework by Grilo et al. is discussed in depth in a previous paper, and is based on the social ecological approach, it needs to be described further in the Introduction. What are the specific ways it expands on, or draws from, Bronfenbrenner’s approach? One question I had that is relevant to the current article: Why is identity based stress a causal influence on identification – is it not plausible that identification styles and how they are perceived can impact identity based stress (e.g., through identity invalidation)? What part(s) of this model is the current research drawing from or seeking to document?

2. Operationalization and measurement of the life course aspect needs further clarification.

2a. Saying that the current study looks at influences on identification “throughout the life course” seems a bit overstated given such a restrictive age range (18-29). I recommend adjusting this language to be more reflective that the authors are studying emerging adults and adults. In the Results section it becomes obvious that participants were specifically instructed to retrospectively talk about their childhood. Additional detail with the interview prompts may help illuminate how the researchers are thinking about life course factors early on, or perhaps clarifying in the Introduction that the research will examine participants’ experiences from childhood to present.

2b. As another example of needing clarification on life course approaches, the Results section states, “These influential interactions were regarded as early, yet continuous, influences on their racial and ethnic identity” (pg. 17). Yet, none of the quotes in this section discuss identification in this way – most of the quotes are in the current tense. The summary on this section seems more tied to the themes in the “Life Course” section below it.

3. The authors discuss health and well-being at length in the abstract, introduction, and discussion, and even label it as a topic within the semi-structured interview. The word health never appears in the Results. Although I may have missed it, none of the quotes discuss health directly. There are references to feeling upset, or belonging, but most of the data center around identity development.

4. There is insufficient detail regarding the semi-structured interview. Who conducted the interviews? What were the semi-structured prompts? In what ways were interviewers instructed to adapt or expand on certain topics?

Some additional points:

- A few typos throughout the paper (e.g., page 1 “Understanding Using this framework”)

- Multi-methods are usually used to refer to approaches that combine multiple methods – from my reading of the manuscript, qualitative analyses of interviews were used. Can the authors expand on how this is mult-method research or remove that claim from the abstract?

- The Discussion should include limitations of a top-down approach to the qualitative analysis (i.e., imposing a framework on the data).

To summarize, the themes were well described with supporting data. However, the paper’s Introduction, Discussion and Results are disjointed. For example, the Intro seems to emphasize identity development, structural factors, and influences on health. The Discussion emphasizes life course differences. Overall, the research would benefit from more clarification about the aims, how the described framework informed the current aims, and more detail regarding the methodological decisions. With revision and more transparency on the interviews and data analysis approach, the contribution could be improved.

Reviewer #2: I want to thank the author(s) for this incredible undertaking and the exploration of multiracial identity formation (from a retrospective point of view) and their proposition of a framework to understand multiracial identity development. I appreciate that the author(s) took a qualitative approach with many interviews that were also geographically varied.

There are a number of positive aspects related to the current manuscript. I am providing some suggestions to the author(s) with the hope that they might further refine their paper.

The author(s) are examining multiracial identity development and its evolution over the life course, and so I would like to see more focus in the literature review that addresses how racialization of people of Color can often create psychological conflict and tension. That is, what the individual internally subscribes to as an identity and culture is not necessarily how the world will treat them because society ascribes an identity to them – essentially racializing them based on physical features and anything that can differentiate them from being white. Often this racialization against the individual is to the “lowest” racial group.

Toward this idea, I think it would be important to talk about racial trauma and how that can be intergenerational (from parent/caregiver) to child and something that the child holds onto across their lifespan. I wonder if the author(s) asked about this. If not, why?

How was the interview questions developed? Did the author(s) develop the interview items from something else or run the questions by any multiracial individuals?

For the quotes, while extensive, they could be trimmed down and made more concise. I would like to see a bit more concision related to the quotes and a bit more on how to make sense of the quotes. That is, how might we interpret the quotes from the participants?

I would like to see more on the methodology, the thematic analyses. Citations? Perhaps a reference to a previous study using the same method.

6. PLOS authors have the option to publish the peer review history of their article (what does this mean?). If published, this will include your full peer review and any attached files.

Reviewer #1: **Yes: **Jasmine Norman

Reviewer #2: **Yes: **William Ming Liu, Ph.D.

---

## [Author Response · Author response to Decision Letter 0]

3 Feb 2023

Please see attached response to reviewers.

---

## [Decision Letter · Decision Letter 1]

28 Mar 2023

The sum of all parts: A multi-level exploration of racial and ethnic identity formation during emerging adulthood

PONE-D-22-12004R1

Dear Dr. Grilo,

We’re pleased to inform you that your manuscript has been judged scientifically suitable for publication and will be formally accepted for publication once it meets all outstanding technical requirements.

Kind regards,

Pieter-Paul Verhaeghe

Academic Editor

PLOS ONE

Additional Editor Comments (optional):

Reviewers' comments:

Reviewer's Responses to Questions

**Comments to the Author**

1. If the authors have adequately addressed your comments raised in a previous round of review and you feel that this manuscript is now acceptable for publication, you may indicate that here to bypass the “Comments to the Author” section, enter your conflict of interest statement in the “Confidential to Editor” section, and submit your "Accept" recommendation.

Reviewer #2: All comments have been addressed

2. Is the manuscript technically sound, and do the data support the conclusions?

Reviewer #2: Yes

3. Has the statistical analysis been performed appropriately and rigorously? 

Reviewer #2: N/A

4. Have the authors made all data underlying the findings in their manuscript fully available?

Reviewer #2: Yes

5. Is the manuscript presented in an intelligible fashion and written in standard English?

Reviewer #2: Yes

6. Review Comments to the Author

Reviewer #2: PONE-D-22-12004R1

The sum of all parts: A multi-level exploration of racial and ethnic identity formation during emerging adulthood

I was one of the original reviewers for this manuscript. In reading the revisions and the manner in which the author(s) attended to my comments/suggestions, I believe that the author(s) have done well in satisfying my recommendations. The paper reads much better and talks more to the issue of racialization. The author(s) trimmed the quotations and made them more specific, and thus, the results section is more concise. I appreciate that the author(s) considered the racial trauma aspects and I understand why they could not include it. I hope they may consider it for a future study since this may connect current experiences of racism with their life-long traumas.

7. PLOS authors have the option to publish the peer review history of their article (what does this mean?). If published, this will include your full peer review and any attached files.

Reviewer #2: **Yes: **William Ming Liu

---

## [Editor Report · Acceptance letter]

31 Mar 2023

PONE-D-22-12004R1 

The sum of all parts: A multi-level exploration of racial and ethnic identity formation during emerging adulthood 

Dear Dr. Grilo:

I'm pleased to inform you that your manuscript has been deemed suitable for publication in PLOS ONE. Congratulations! Your manuscript is now with our production department. 

Kind regards, 

on behalf of

Professor Pieter-Paul Verhaeghe 

Academic Editor

PLOS ONE